# UNDERSTANDING PATHOLOGIES OF DEEP HETEROSKEDASTIC REGRESSION

## ABSTRACT

Several recent studies have reported negative results when using heteroskedastic neural regression models to model real-world data. In particular, for overparameterized models, the mean and variance networks are powerful enough to either fit every single data point (while shrinking the predicted variances to zero), or to learn a constant prediction with an output variance exactly matching every predicted residual (i.e., explaining the targets as pure noise). This paper studies these difficulties from the perspective of statistical physics. We show that the observed instabilities are not specific to any neural network architecture but are already present in a *field theory* of an overparameterized conditional Gaussian likelihood model. Under light assumptions, we derive a *nonparametric free energy* that can be solved numerically. The resulting solutions show excellent qualitative agreement with empirical model fits on real-world data and, in particular, prove the existence of *phase transitions*, i.e., abrupt, qualitative differences in the behaviors of the regressors upon varying the regularization strengths on the two networks. Our work thus provides a theoretical explanation for the necessity to carefully regularize heteroskedastic regression models. Moreover, the insights from our theory suggest a scheme for optimizing this regularization which is quadratically more efficient than the naïve approach.

## 1 INTRODUCTION

Regression and classification problems lie at the heart of deep learning (Mathew et al., 2021; Ahmad et al., 2019; Krizhevsky et al., 2012). Homoskedastic regression models assume constant (e.g., Gaussian) output noise and mainly amount to learning a function $f(x)$ that tries to predict the most likely target $y$ for input $x$. In contrast, heteroskedastic models assume that the output noise may depend on the input features $x$ as well, and try to learn a conditional distribution $p(y|x)$ with non-uniform variance. This not only allows the model to assign different importances to training data, but ultimately results in a model that "knows where it fails" (Skafte et al., 2019; Fortuin et al., 2022).

Unfortunately, learning overparameterized heteroskedastic regression models (e.g., using deep neural networks) has proven to be difficult, as these models are prone to extreme forms of overfitting (Lakshminarayanan et al., 2017; Nix & Weigend, 1994). On the one hand, the mean model is flexible enough to perfectly fit every training datum's target, while the standard deviation network learns to maximize the likelihood by shrinking the predicted standard deviations to zero. On the other hand, with the incorrect strength of regularization on the mean network's parameters will make it prefer a constant solution. Such a flat prediction can be accomplished by allowing the standard deviation network to explain all residuals as random noise, thus overfitting to the empirical prediction residuals of the data. Both types of overfitting are shown in Fig. 1.

While several practical solutions to learning overparameterized heteroskedastic regression models have been proposed (Skafte et al., 2019; Stirn & Knowles, 2020; Seitzer et al., 2022; Stirn et al., 2023), no comprehensive theoretical study of the failure of these methods has been offered so far. We conjecture this is because overparameterized models have attracted the most attention only in the past few years, while most classical statistics have focused on under-parameterized (e.g., linear) regression models where such problems cannot occur (Huber, 1967; Astivia & Zumbo, 2019).

This paper provides a theoretical analysis of the failure of heteroskedastic regression models in the overparameterized limit. To this end, it borrows a tool that abstracts away from any details of the

involved neural network architectures: classical field theory from statistical mechanics (Landau & Lifshitz, 2013; Altland & Simons, 2010). Via our field-theoretical description, we can recover the optimized heteroskedastic regressors as solutions to partial differential equations that can be derived from a variational principle. These equations can in turn be solved numerically by optimizing the field theory's free energy functional.

Our analysis results in a two-dimensional *phase diagram*, representing the coarse-grained behavior of heteroskedastic noise models for every parameter configuration. Each of the two dimensions corresponds to a different level of regularization of either the mean or standard deviation network. As encountered in many complex physical systems, the field theory unveils *phase transitions*, i.e., sudden and discontinuous changes in certain *observables* (metrics of interest) that characterize the model, such as the smoothness of its mean prediction network, upon small changes in the regularization strengths. In particular, we find a sharp phase boundary between the two types of behavior outlined in the first paragraph, at weak regularization.

Our contributions are as follows:

• We provide a unified theoretical description of overparameterized heteroskedastic regression models, which generalizes across different models and architectures, drawing on tools from statistical mechanics and variational calculus.

• In this framework, we derive a field-theoretical nonparametric free energy (NFE), which can explain the observed types of overfitting in these models and describe *phase transitions* between them.

• Empirically, we show excellent qualitative agreement of our NFE with experiments, both on simulated and real-world regression tasks.

• We find evidence that a one-dimensional search over hyperparameters is sufficient to achieve well-calibrated models instead of a comprehensive search over the entire two-dimensional *phase diagram*.

## 2 A FIELD THEORY FOR OVERPARAMETERIZED HETEROSKEDASTIC REGRESSION

**Heteroskedastic Regression**    Consider the setting in which we have a collection of independent data points $\mathcal{D} := \{(x_i, y_i)\}_{i=1}^N$ with covariates $x_i \in \mathcal{X} \subset \mathbb{R}^d$ drawn from some distribution $x_i \sim p(x)$ and response values $y_i \in \mathcal{Y} \equiv \mathbb{R}$ normally distributed with unique mean $\mu_i$ and precision (inverse-variance) $\Lambda_i > 0$ (i.e., $y_i \sim \mathcal{N}(\mu_i, \Lambda_i)$). We assume to be in a *heteroskedastic* setting, in which $\Lambda_i$ need not equal $\Lambda_j$ for $i \neq j$. Finally, we assume *both* the mean and standard deviation of $y_i$ to be explainable via $x_i$:

$$y_i \,|\, x_i \sim \mathcal{N}(\mu(x_i), \Lambda(x_i)^{-\frac{1}{2}}) \text{ for } i = 1, \ldots, N \tag{1}$$

with continuous functions $\mu : \mathcal{X} \to \mathbb{R}$ and $\Lambda : \mathcal{X} \to \mathbb{R}_{>0}$. In a modeling setting, learning $\Lambda$ can be interpreted as directly estimating and quantifying the *aleatoric* (data) uncertainty.

**Overparameterized Neural Networks**    There exist many options for modeling $\mu$ and $\Lambda$. Of particular interest to many is representing each of these functions as neural networks—specifically ones that are overparameterized. These models are well-known *universal function approximators*, which makes them great choices for estimating the true functions $\mu$ and $\Lambda$ (Hornik, 1991).

Let the mean network $\hat{\mu}_\theta : \mathcal{X} \to \mathbb{R}$ and standard deviation network $\hat{\Lambda}_\phi : \mathcal{X} \to \mathbb{R}_{>0}$ be arbitrary depth, overparameterized feed-forward neural networks parameterized by $\theta$ and $\phi$ respectively. For a given input $x_i$, these networks collectively represent a corresponding predictive distribution for $y_i$:

$$\hat{p}(y_i \,|\, x_i) := \mathcal{N}(y_i; \hat{\mu}_\theta(x_i), \hat{\Lambda}_\phi(x_i)^{-\frac{1}{2}}). \tag{2}$$

**Pitfalls of MLE**    Our assumed form of data naturally suggests training $\hat{\mu}_\theta$ and $\hat{\Lambda}_\phi$, or rather learning $\theta$ and $\phi$, by minimizing the cross-entropy between the joint data distribution $p := p(x, y) = p(y \,|\, x)p(x)$ and the induced predictive distribution $\hat{p} := \hat{p}(y \,|\, x)p(x)$. This objective is defined as

$$\mathcal{L}(\theta, \phi) := H(p, \hat{p}) = -\mathbb{E}_p \left[ \log \hat{p}(x, y) \right] \tag{3}$$

$$= \int_{\mathcal{X}} p(x) \int_{\mathcal{Y}} p(y \,|\, x) \log \mathcal{N}(y; \hat{\mu}_\theta(x), \hat{\Lambda}_\phi(x)^{-\frac{1}{2}}) dy dx + c$$

where $c$ is a constant with respect to $\theta$ and $\phi$. This expectation is often approximated using a Monte Carlo (MC) estimate with $N$ samples, yielding the following tractable objective function:

$$\mathcal{L}(\theta, \phi) \approx \frac{1}{2N} \sum_{i=1}^{N} \hat{\Lambda}_\phi(x_i)(y_i - \hat{\mu}_\theta(x_i))^2 - \log \hat{\Lambda}_\phi(x_i). \tag{4}$$

Minimizing this cross-entropy objective function with respect to parameters $\theta$ and $\phi$ using data samples is synonymous with maximum likelihood estimation (MLE).

Unfortunately, given an infinitely flexible model, this objective function is ill-posed for our purposes. Consider the fact that there is implicit regularization for $\phi$, as the first term in Eq. (4) is minimized when $\hat{\Lambda}_\phi \to 0$, while the second term is minimized when $\hat{\Lambda}_\phi \to \infty$. This alone would be fine; however, the mean function $\hat{\mu}_\theta$ has no such regularization, so inevitably during training it will estimate $y$ perfectly (or rather to arbitrary precision) for at least a single data point $(x_i, y_i)$. Once this happens, the residual for this input, $y_i - \hat{\mu}_\theta(x_i)$, approaches 0 and the regularization for $\hat{\Lambda}_\phi$ vanishes, at least at this point $x_i$. This leads to the predicted precision diverging towards $\infty$. Once training has reached this point, the objective function becomes completely unstable due to effectively containing a term whose limit naïvely yields $\infty - \infty$.[1]

**Regularization** Even though $\hat{\Lambda}_\phi$ is implicitly regularized in the standard cross-entropy loss as mentioned earlier, we posit that additional regularization on $\hat{\Lambda}_\phi$, or rather $\phi$, is required to avoid this instability. It can be tempting to think that one must regularize $\theta$ in order to avoid overfitting. And while this is generally true, the objective function $\mathcal{L}$ will still be unstable so long as *at least* one input $x_i$ yields a perfect prediction (*i.e.*, $y_i = \hat{\mu}_\theta(x_i)$). This situation is still fairly likely to occur even in the most regularized mean predictors and cannot be avoided, especially if $\{y_i\}$ is zero-centered.

To prevent this from happening, we can include $L_2$ penalty terms for both $\theta$ and $\phi$ in our loss function:

$$\mathcal{L}_{\alpha,\beta}(\theta, \phi) := \mathcal{L}(\theta, \phi) + \alpha||\theta||_2^2 + \beta||\phi||_2^2 \tag{5}$$

where $\alpha, \beta > 0$ are penalty coefficients. Intuitively, the primary role of regularizing $\theta$ is to avoid mean prediction overfitting while the role of regularizing $\phi$ is to provide stability and control complexity in the predicted aleatoric uncertainty. As $\alpha \to \infty$, the network models a constant mean and as $\beta \to \infty$ we effectively model a homoskedastic regime.[2]

**Reparameterizing the Regularization** We introduce an alternative parameterization of the regularization coefficients:

$$\mathcal{L}_{\rho,\gamma}(\theta, \phi) := \rho\mathcal{L}(\theta, \phi) + (1 - \rho)\left[\gamma||\theta||_2^2 + (1 - \gamma)||\phi||_2^2\right] \tag{6}$$

and we restrict $\rho, \gamma \in (0, 1)$. This parameterization is one-to-one with the $\alpha, \beta$ parameterization (with $\alpha = \frac{(1-\rho)\gamma}{\rho}$ and $\beta = \frac{(1-\rho)(1-\gamma)}{\rho}$) and it can be shown that $\nabla_{\theta,\phi}\mathcal{L}_{\rho,\gamma} \propto \nabla_{\theta,\phi}\mathcal{L}_{\alpha,\beta}$, thus minimizing one objective is equivalent to minimizing the other. Because $\rho, \gamma$ are bounded we are able to completely cover the space of regularization combinations by searching over $(0, 1)$ whereas in the $\alpha, \beta$ parameterization $\alpha, \beta \in \mathbb{R}_{>0}$ are unbounded. Now, $\rho$ determines the relative importance between the likelihood and the total regularization imposed on both networks. On the other hand, $\gamma$ weights the proportion of total regularization between the mean and precision networks. Here, $\rho = 1$ corresponds to the MLE objective while $\rho \to 0$ could be interpreted as converging to the mode of the prior in a Bayesian setting. Fixing $\gamma = 1$ leads to an unregularized precision function while choosing $\gamma = 0$ results in an unregularized mean function.

**Qualitative Description of Phases** Model solutions across the space of $\rho$ and $\gamma$ hyperparameters exhibit different traits and behaviors. Similar to physical systems, this can be described as a collection of typical states or *phases* that make up a *phase diagram* as a whole. These phase diagrams are

---

[1]Note that this is predicated on the model being flexible enough to allow for large changes in predictions $\hat{\mu}_\theta(x)$ and $\hat{\Lambda}_\phi(x)$ after iteratively updating parameters $\theta$ and $\phi$ while allowing for minimal changes in neighboring predictions (i.e., $\hat{\mu}_\theta(x')$ and $\hat{\Lambda}_\phi(x')$ for some $x' \in \mathcal{X}$ such that $0 < ||x - x'|| < \epsilon$).

[2]This is under the assumption that either the networks have an unpenalized bias term in the final layer *or* that the data is standardized to have zero mean and unit variance.

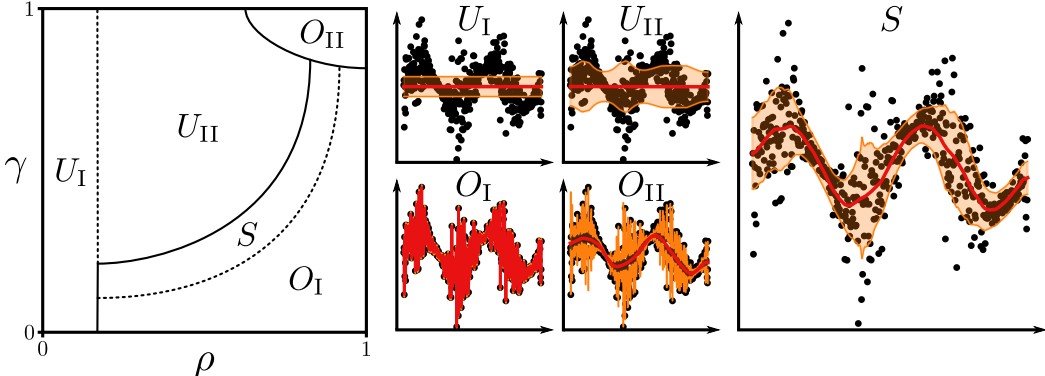

Figure 1: Visualization of a typical phase diagram in $\rho - \gamma$ regularization space for a heteroskedastic regression model shown on left. Solid and dotted lines indicate sharp and smooth transitions in model behavior respectively. Example model mean fits shown in red (with pointwise $\pm$ standard deviation in orange) from the NFE for each key phase in middle and right plots.

typically consistent in shape across datasets and methodologies. Fig. 1 shows an example phase diagram along with model fits coming from specific $(\rho, \gamma)$ pairings. We argue that there are five primary regions of interest:

• Region $U_{\mathrm{I}}$: Both the mean and precision functions are heavily regularized. In this region, the likelihood is so lowly weighted it is as if the model had not seen the data. Regardless of the prioritization ($\gamma$ value) of mean versus precision function, the likelihood plays a minor role in the objective. The mean function is a constant through zero while the standard deviation is fixed on 1 (the values they were initialized to).

• Region $U_{\mathrm{II}}$: In this region the mean function is still heavily regularized and tends to be flat, underfitting the data as in Region $U_I$. However, the strength of regularization comes from a more even combination of both $\rho, \gamma$. Despite the constant behavior of the mean function, the precision function can still accommodate the residuals and the prediction intervals adapt with the data.

• Region $O_{\mathrm{I}}$: Heavily overfit mean and the residuals and corresponding standard deviations essentially vanish. Increasing $\rho \to 1$ yields true MLE fits and this is seen on the right. This portion of the phase exists across a wide range of $\gamma$ values. Low values of $\gamma$ restrict the flexibility of the precision function, but due to the overfitting in the mean, the flexibility is not needed to fit the residuals.

• Region $O_{\mathrm{II}}$: In this region the mean function does not overfit due to regularization, leaving large residuals for the lowly regularized precision function to overfit onto.

• Region $S$: Here, the model is well calibrated—the mean function and standard deviations adapt to the data without overfitting. We conjecture that solutions in this region will provide the best generalization.

**Nonparametric Modeling & Field Theory**  While we have a somewhat intuitive grasp over the effects of the regularization hyperparameters on the learned model, directly analyzing this behavior with neural models can be untrustworthy due to requiring potentially unstable optimization techniques and dealing with possible identifiability issues and local minima. As such, we propose to create an investigatory tool that will assist in more directly analyzing these behaviors. Firstly, we propose abstracting the neural networks $\hat{\mu}_\theta$ and $\hat{\Lambda}_\phi$ with nonparametric, twice-differentiable functions $\hat{\mu}$ and $\hat{\Lambda}$ respectively. Since these functions are nonparametric, we can no longer use $L_2$ penalties. A somewhat comparable substitute is to directly penalize the output "complexity" of the models, or in other words the cumulative absolute rate of change: $\int \alpha ||\nabla \hat{\mu}(x)||_2^2 dx$ and $\int \beta ||\nabla \hat{\Lambda}(x)||_2^2 dx$. Note that these specific penalizations induce similar limiting behaviors for resulting solutions (i.e., $\alpha, \beta \to 0$ implies overfitting while $\to \infty$ implies constant functions). In the case where $\hat{\mu}_\theta$ and $\hat{\Lambda}_\phi$ are linear models, this squared gradient penalty is equivalent to a $L_2$ penalty on the weights.

*Field theories* are non-parametric descriptions of the spatial (or spatiotemporal) configurations of continuous physical systems (Altland & Simons, 2010). For example, the local magnetic field

(magnetization) of a two or three-dimensional magnetic material would be an example of a *field*. For time independent problems we minimize a free energy functional: we will refer to it as *nonparametric free energy (NFE)*. The NFE depends on certain parameters, such as the strength of an external magnetic field or a temperature. Upon smoothly varying these parameters, the most likely field configuration can undergo smooth changes ("crossovers") or abrupt changes ("phase transitions"). As follows, we outline a field-theoretical treatment of the mean and variance parameters of our considered heteroskedastic regression model, where the mean and variance show phase transitions upon varying their regularization strengths.

Using the assumptions outlined above, the cross-entropy objective can be interpreted as an action functional of a corresponding two-dimensional NFE:

$$\mathcal{L}_{\rho,\gamma}(\hat{\mu}, \hat{\Lambda}) = \int_{\mathcal{X}} p(x)\rho \int_{\mathcal{Y}} p(y\,|\,x) \left[ \frac{1}{2}\hat{\Lambda}(x)\,(y - \hat{\mu}(x))^2 - \frac{1}{2}\log\hat{\Lambda}(x) \right] dy$$
$$+ (1 - \rho)\left[ \gamma||\nabla\hat{\mu}(x)||_2^2 + (1 - \gamma)||\nabla\hat{\Lambda}(x)||_2^2 \right] dx. \tag{7}$$

While this is the full NFE, it is cumbersome to analyze due to the nested integral. As such, we consider the scenario in which the inner integral is approximated using a single MC sample. As this must be done for every $x \in \mathcal{X}$, we aggregate all of these samples into an indexed set $y(\cdot) := \{y(x)\}_{x\in\mathcal{X}}$ where $y(x) \sim p(y\,|\,x)$. One can view this collection as a stochastic process (specifically a white noise process scaled by true precision $\Lambda(x)$ and shifted by true mean $\mu(x)$). This approximation yields the following simplified NFE:

$$\mathcal{L}_{\rho,\gamma}(\hat{\mu}, \hat{\Lambda}) \approx \int_{\mathcal{X}} p(x)\rho \left[ \frac{1}{2}\hat{\Lambda}(x)\,(y(x) - \hat{\mu}(x))^2 - \frac{1}{2}\log\hat{\Lambda}(x) \right]$$
$$+ (1 - \rho)\left[ \gamma||\nabla\hat{\mu}(x)||_2^2 + (1 - \gamma)||\nabla\hat{\Lambda}(x)||_2^2 \right] dx. \tag{8}$$

We are primarily interested in solutions $\hat{\mu}^*$ and $\hat{\Lambda}^*$ that minimize the NFE $\mathcal{L}_{\rho,\gamma}(\hat{\mu}, \hat{\Lambda})$ as these are roughly analogous to models $\hat{\mu}_\theta$ and $\hat{\Lambda}_\phi$ that minimize penalized cross-entropy $\mathcal{L}_{\rho,\gamma}(\theta, \phi)$. We can gain insights into these solutions by taking functional derivatives of the NFE with respect to $\hat{\mu}$ and $\hat{\Lambda}$ and setting them to zero. We arrive at the following conditions that must simultaneously be met for any set of solutions:

$$\begin{cases} \frac{\delta\mathcal{L}_{\rho,\gamma}(\hat{\mu},\hat{\Lambda})}{\delta\hat{\mu}} \overset{\triangle}{=} 0 \\ \frac{\delta\mathcal{L}_{\rho,\gamma}(\hat{\mu},\hat{\Lambda})}{\delta\hat{\Lambda}} \overset{\triangle}{=} 0 \end{cases} \implies \begin{cases} \hat{\Lambda}^*(x)(\hat{\mu}^*(x) - y(x)) = 2\left(\frac{1-\rho}{\rho}\right)\gamma\frac{\Delta\hat{\mu}^*(x)}{p(x)} \\ (\hat{\mu}^*(x) - y(x))^2 = \frac{1}{\hat{\Lambda}^*(x)} + 4\left(\frac{1-\rho}{\rho}\right)(1 - \gamma)\frac{\Delta\hat{\Lambda}^*(x)}{p(x)}, \end{cases} \tag{9}$$

where $\Delta$ is the Laplace operator (Engel & Dreizler, 2011). Note that these equalities hold true *almost everywhere* (a.e.) with respect to $p(x)$. Interestingly, both resulting relationships include a regularization coefficient divided by the density of $x$. This makes intuitive sense as while the functions as a whole have a global level of regularization dictated by $\rho$ or $\gamma$, locally this regularization strength is augmented proportional to how unlikely the input is. This will lead to high-density regions of $x$ allowing for more complexity while forcing less likely regions to produce simpler, less potentially erratic outputs. Similarly, we can glean that $\rho$ and $\gamma$ directly impact the complexity of $\hat{\mu}$ and $\hat{\Lambda}$ by scaling the importance of the *curvature* of these functions, or rather $\Delta\hat{\mu}$ and $\Delta\hat{\Lambda}$.

**Numerically Solving the NFE** In practice we discretize the NFE to arrive at approximate solutions. Let $\{x_i\}_{i=1}^{n_{ft}}$ be a set of fixed points in $\mathcal{X}$ that we assume are evenly spaced. Define $\vec{\mu}, \vec{\Lambda}, \vec{y}$ to be $n_{ft}$-dimensional vectors where for each $i$, $\vec{\mu}_i = \hat{\mu}(x_i), \vec{\Lambda}_i = \hat{\Lambda}(x_i), y_i = y(x_i)$. We solve for the optimal $\vec{\mu}$ and $\vec{\Lambda}$ using the discretized approximation to Eq. (7) via gradient based optimization methods:

$$\mathcal{L}_{\rho,\gamma}(\vec{\mu}, \vec{\Lambda}) \approx \sum_{i=1}^{n_{ft}} \rho\left[ \frac{1}{2}\vec{\Lambda}_i\,(y_i - \vec{\mu}_i)^2 - \frac{1}{2}\log\vec{\Lambda}_i \right] + (1 - \rho)\left[ \gamma||\nabla\vec{\mu}_i||_2^2 + (1 - \gamma)||\nabla\vec{\Lambda}_i||_2^2 \right] \tag{10}$$

and the gradients of $\hat{\mu}, \hat{\Lambda}$ are approximated numerically (Fornberg, 1988).

Table 1: NFE Limiting Cases. We provide intuition for the consequences of Prop. 1 and match the limits to the phase diagram regions in Fig. 1.

| Regularization | Outcome |
| --- | --- |
| $\rho \to 1, \gamma \in [0, 1]$ | This is equivalent to MLE. Approaching $\rho = 1$, we observe overfit mean solutions. According to the conditions in Eq. (9) this holds for any value of $\gamma$. We see this in regions $O_\mathrm{I}$ and $O_\mathrm{II}$ in Fig. 1. In theory, at $\rho = 1$, there is a contradiction implying no solution should exist. |
| $\rho \to 0, \gamma \in (0, 1)$ | This setting places all the weight of the objective on the regularizers, completely ignoring the data. This corresponds with region $U_\mathrm{I}$. In theory, the optimal solution at $\rho = 0$ is for both $\hat{\mu}, \hat{\Lambda}$ to be constant (flat) functions. |
| $\rho \in (0, 1), \gamma \to 1$ | All of the regularization is placed on the mean function, leading to mean underfitting. However, the precision is unregularized and $\Lambda^{-1/2}$ thus matches the residuals perfectly. This is the top row of the phase diagrams. |
| $\rho \in (0, 1), \gamma \to 0$ | The mean is unregularized and the precision is strongly regularized. These fits are characterized by severe overfitting and can be found along the bottom row of the phase diagrams. |

**NFE Insights** The pair of constraints in Eq. (9) allow us to glean useful insights into the resulting regularized solutions by looking at edge cases of specific combinations of $\rho$ and $\gamma$ values. We summarize the theoretical properties of the limiting cases of $\rho, \gamma$ approaching extreme values in the proposition below and in Table 1. Please refer to Appendix A.2 for the proofs to these claims.

**Proposition 1.** *Under the assumptions of our NFE (see above), the following properties hold: (i) in the absence of regularization ($\rho = 1$), there are no solutions to the NFE; (ii) in the absence of data ($\rho = 0$), there is no unique solution to the NFE; and (iii) there are no valid solutions to the NFE if $\rho \in (0, 1)$ and $\gamma = 1$ (should there be no mean regularization, then there needs to be at least some regularization for the precision).*

Importantly, these limiting cases match our intuition for the solutions to the NFE (which were equivalent to the neural network setting). Furthermore, if we assume that there exist valid solutions for $\gamma, \rho \in (0, 1)$ then it stands to reason that the solutions should either experience a sharp transition or a smooth cross-over between the behaviors described in the limiting cases. Empirically, we have discovered that the phase diagram typically resembles Fig. 1. We leave the analytical justification for the types of boundaries and their shapes/placement in the phase diagram for future work.

## 3 EXPERIMENTS

**Datasets** We analyze the effects of regularization on several one-dimensional simulated datasets and standardized versions of the *Concrete* (Yeh, 2007), *Housing* (Harrison & Rubinfeld, 1978), *Power* (Tüfekci, 2014), and *Yacht* (Gerritsma, 1981) regression datasets from the UC Irvine Machine Learning Repository (Kelly et al.). We fit neural networks to the simulated and real-world data and additionally solve our NFE for the simulated data. Detailed descriptions of the data are included in Appendix B.1. We present the results for a simulated sinusoidal (*Sine*) dataset as well as the four UCI regression datasets and have results for the other simulated datasets in Appendix B.5.

**Modeling Choices** We chose $\hat{\mu}_\theta, \hat{\Lambda}_\phi$ to be fully-connected networks with three hidden layers of 128 nodes and leaky ReLU activation functions. The first half of training was only spent on fitting $\hat{\mu}_\theta$, while in the second half of training, both $\hat{\mu}_\theta$ and $\hat{\Lambda}_\phi$ were jointly learned. This improves stability, since the precision is a function of the mean $\hat{\mu}_\theta$, and is similar in spirit to ideas presented in (Skafte et al., 2019). Complete training details can be found in Appendix B.2.

**Metrics of Interest** We consider two metrics of interest in our experiments. Firstly, the *Sobolev norm*, $\int_{\mathcal{X}} ||\nabla f(x)||_2^2 \, dx$, where $f$ is one of the learned $\hat{\mu}_\theta, \hat{\Lambda}_\phi, \vec{\mu}, \vec{\Lambda}$. It captures how expressive a

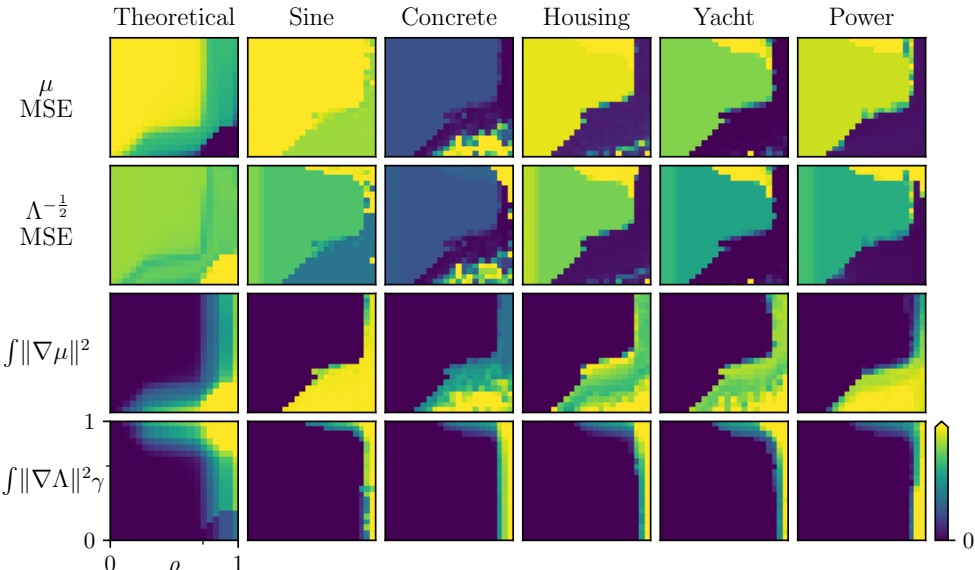

Figure 2: Array plot of various metrics (rows) evaluated on different data or fitting techniques (columns). The left most column holds the results from the theoretical NFE that we developed. The remaining columns show results from fitting neural networks to data. (Data sets refer to test splits.) The values of the six runs are averaged. The intermediate ticks on the lower left plot mark $\gamma = 0.5$ and $\rho = 0.5$. The shapes of the regions and transitions between regions are of particular interest. Our NFE matches the empirical phase diagrams well in this regard, and the phase transitions are preserved across different models and datasets.

learned function is, with more expressive functions yielding a higher value. Secondly, we consider the mean squared error (MSE). We measure this quantity between predicted mean $\hat{\mu}_\theta(x_i)$ and target $y_i$, as well as between predicted standard deviation ($\Lambda^{-1/2}(x_i)$) and absolute residual $|\hat{\mu}_\theta(x_i) - y_i|$. If the mean and standard deviation are well-fit to the data, both of these values should be low. We opt for $\Lambda^{-\frac{1}{2}}$ MSE due to its similarities to variance calibration (Skafte et al., 2019) and expected normalized calibration error (Levi et al., 2022). It should be noted that we use this to measure our uncertainty quantification over other calibration metrics, such as expected calibration error (ECE), as they have been shown to give good scores in degenerate cases (Kuleshov et al., 2018; Chung & Neiswanger, 2021; Levi et al., 2022).[3]

**Plot Interpretation** We present summaries of the fitted models in grids with $\rho$ on the $x$-axis and $\gamma$ on the $y$-axis in Fig. 2. The far right column ($\gamma = 1$) corresponds to MLE solutions. The main focus is on qualitative traits of fits under different levels of regularization and how they behave in a relative sense, rather than a focus on absolute values. Fig. 3 show the summary statistics along the slice where $\rho = 1 - \gamma$. Zero on these plots corresponds to the upper left-hand corner while one corresponds to the lower right-hand corner.

**Observation 1:** *Our metrics show sharp phase transitions upon varying $\rho, \gamma$, as in a physical system.* Fig. 2 and Fig. 3 show a sharp transition, both leading to worsening and improving performance when moving along the minor diagonal. In totality, across the four metrics, the five regions are apparent. But not all of the regions in Fig. 1 appear in the heatmaps of each metric. For example, region $O_{II}$ does not always appear in the quantities related to the mean. When using neural networks to approximate $\mu$ and $\Lambda$, there are sharper boundaries between phases than in the NFE's numerical solutions. The boundary between $U_{II}$ and $O_I$ is sharply observed in the plots of $\int ||\nabla\mu(x)||_2^2 \, dx$. However, in terms of $\mu$ MSE, a smoother transition (i.e., the $S$ region) is visible.

---

[3]Namely, for ECE, a model can achieve a low error by learning marginal statistics of the dataset and effectively ignore any trends present between $x$ and $y$. We would prefer our uncertainty quantification metric to help differentiate between $U_I, U_{II}$, and $O_{II}$ in our phase diagram.

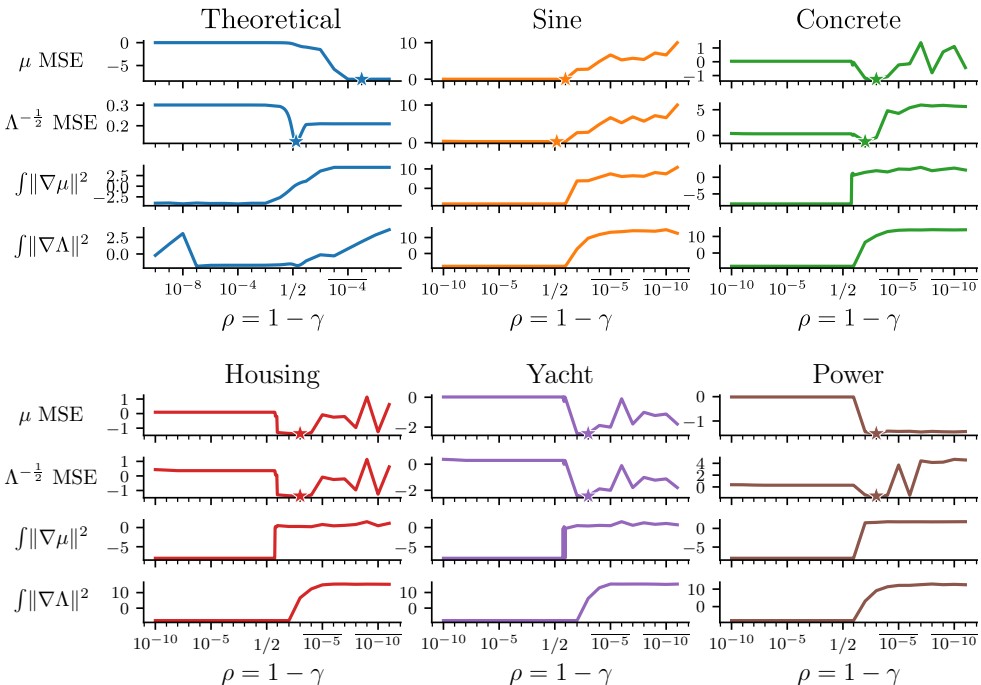

Figure 3: Test metrics for six different settings achieved with varying values of $\rho \in (0, 1)$ and with $\gamma$ restricted to be equal to $1 - \rho$. Stars indicate minimum MSE values. All metrics are reported on a $\log_{10}$ scale. $\rho$ values are shown on a logit scale with $\overline{10^k} := 1 - 10^k$. From left to right, note the sharp decrease in test metric values, especially in the solutions to neural network models followed by a typical smoother increase. This empirically supports the existence of the well-calibrated $S$ phase shown in Fig. 1 and allows for hyperparameter optimization in $O(N)$ instead of $O(N^2)$.

**Observation 2:** *The NFE insights and observed phases are consistent with the numerically solved non-parametric NFE and the results from fitting neural networks. Thus, our results are not tied to a specific architecture or dataset.* In alignment with our theoretical insights, phases $U_I$ and $O_I$ exhibit consistent behavior across values of $\gamma$ (vertical slices in the phase diagrams in Fig. 2). In the right-hand columns ($\rho \to 1$), there is near-perfect matching of the data by the mean function and this is also visible in the lower rows ($\gamma \to 0$). Within the metrics we assess, the shapes of the regions vary with the level of regularization in a similar fashion on all datasets. In the plots of $\int ||\nabla \Lambda(x)||_2^2 \, dx$, the region where $\Lambda$ is flatter covers a larger area compared to the phase diagram showing $\int ||\nabla \mu(x)||_2^2 \, dx$. That is, for the same proportion of regularization as the mean, the precision remains flatter.

**Observation 3:** *We can search along $\rho = 1 - \gamma$ to find a well-calibrated $(\rho, \gamma)$-pair from region $S$.* Our NFE predicts that a slice across the minor diagonal of the phase diagram should always cross the $S$ region (see Fig. 1). Fig. 3 show that by searching along this diagonal, we indeed find a combination of regularization strengths where both mean and standard deviation generalize well to held-out test data. This implies that there is no need to search the entirety of the two-dimensional space, but only a single slice which reduces the number of models to fit from $O(N^2)$ to $O(N)$, where $N$ is the number of $\rho$ and $\gamma$ values that are tested. A practical suggestion is to search along $\rho = 1 - \gamma$, i.e., moving from the upper-left to the lower-right corner of the phase diagram. The MSE versions of Fig. 3 show that along this path, the performance is initially poor, improves, and then drops off again. These shifts from strong to weak performance are sharp. The regularization pairings that result in optimal performance with respect to $\mu$- and $\Lambda^{-1/2}$-MSE are close to each other for the real-world test data. As the theory predicts, the performance becomes highly variable as we approach the MLE solutions and the NFE fails to converge in this region. We compare models chosen by our diagonal line search to two heteroskedastic modeling baselines in Appendix D on the synthetic and UCI datasets as well as a scalar quantity from the ClimSim dataset (Yu et al., 2023). In most cases the model chosen via the diagonal line search was competitive or better than the baselines.

## 4 RELATED WORK

Uncertainty can be divided into epistemic (model) and aleatoric (data) uncertainty (Hüllermeier & Waegeman, 2021), the latter of which can be further divided into homoskedastic (constant over the input space) and heteroskedastic (varies over the input space). Handling heteroskedastic noise historically has been and continues to be an active area of research in statistics (Huber, 1967; Eubank & Thomas, 1993; Le et al., 2005; Uyanto, 2022) and machine learning (Abdar et al., 2021), but is less common in deep learning (Kendall & Gal, 2017; Fortuin et al., 2022), probably due to the pathologies that we analyze in this work. Heteroskedastic noise modeling can be interpreted as reweighting the importance of individual datapoints during training time, which Wang et al. (2017) show to be beneficial in the presence of corrupted data and Khosla et al. (2022) in active learning.

To the best of our knowledge, Nix & Weigend (1994) were the first to model a mean and standard deviation function with neural networks and Gaussian likelihood. Seitzer et al. (2022) provide an in-depth analysis of the shortcomings of MLE estimation in this setting and adjust the gradients during train time to avoid pathological behavior. Skafte et al. (2019) suggest changing the optimization loop to train the mean and standard deviation networks separately, treating the standard deviations variationally and integrating them out as Takahashi et al. (2018) does in the context of VAEs, accounting for the location of the data when sampling, and setting a predefined global variance when extrapolating. Stirn & Knowles (2020) also perform amortized VI on the standard deviations and evaluate their model from the perspective of posterior predictive checks. Finally, Stirn et al. (2023) extend the idea of splitting mean and standard deviation network training in a setting where there are several shared layers to learn a representation before emitting mean and standard deviation. While these works propose practical solutions, in contrast to our work, none of them study the theoretical underpinnings of these pathologies, let alone in a model- or data-agnostic way.

## 5 CONCLUSION

We have used field-theoretical tools from statistical physics to derive a nonparametric free energy, which allowed us to produce analytical insights into the pathologies of deep heteroskedastic regression. These insights generalize across models and datasets and provide a theoretical explanation for the need for carefully tuned regularization in these models, due to the presence of sharp phase transitions between pathological solutions. We have also presented a numerical approximation to this theory, which empirically agrees with neural network solutions to synthetic and real-world data. Using insights from the theory, we have shown that we can tune the required regularization for these models more efficiently than would naïvely be the case. Finally, we hope that this work will open an avenue of research for using ideas from theoretical physics to study the behavior of overparameterized models, thus furthering our understanding of otherwise typically unintelligible large models used in AI systems.

**Limitations** Our NFE and subsequent analysis are restricted to regression problems. From an uncertainty quantification perspective, the models we discuss only account for the aleatoric uncertainty. Though our use of regularizers has a Bayesian interpretation, we are not performing Bayesian inference and do not account for epistemic uncertainty. Solving the NFE under a fully Bayesian framework would result in stochastic PDE solutions. We leave analysis of this setting to future work. Additionally, our suggestion to search $\rho = 1 - \gamma$ to find good hyperparameter settings appears to be valid, but it requires fitting many models. Ideally, one might hope to use the field theory directly to find optimal regularization settings for real-world models, but our numerical approximation is currently not accurate enough for this use case.

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
