# OpenReview forum: "Understanding Pathologies of Deep Heteroskedastic Regression"
_ICLR.cc/2024/Conference — Submitted to ICLR 2024_

### Official Review · Reviewer_6L3Q · 2023-10-27

**Soundness:** 3 good
**Presentation:** 3 good
**Contribution:** 2 fair
**Rating:** 5
**Confidence:** 4

**Summary:**

This paper studies learning both the mean and the variance functions using deep neural networks. Estimation of the variance term posts additional difficulty since training may fall into two undesirable scenarios: (1) the inverse variance goes to zero, which means the heteroscedasticity is not learned, or (2) the inverse variance goes to infinity, which means the training data are overfitted.

This paper presents a categorization of the possible scenarios depending on how much regularization is applied to the mean function and variance function. For both mean and variance, there is a potential memorization vs. generalization distinction. In the 2D phase diagram, the interaction of the mean and variance functions results in 5 categories.

Then the authors use heuristic arguments and propose numerical approximation to nonparametric
free energy, which aligns with experiments in relatively simple settings.

**Strengths:**

1. (main) Conceptually interesting: a richer understanding of regularization for both mean and deviation.
2. Uncertain quantification: an important topic, this paper provides some ideas about how to tame overparametrization

**Weaknesses:**

1. (main) Is there a sharp phase transition? This paper lacks quantitative measurement and results. It would be interesting to calculate, at least under certain simple generative models, the free energy and check if there is a first-order/second-order phase transition.
2. (main) Technically speaking, not sure which part of the paper is innovative---for example, Eqns 7--10, are they new or semi-new (i.e., similar derivations are obtained in a different context)?  I would be skeptical that Eqns 7--10 are entirely new. Also, the proposed regularization is well-studied in the literature.
3. Data experiments are simple, but I am mostly fine with that, since this paper is mainly proof-of-concept.
4. It is a bit handwaving when transitioning from parameter norm regularization to gradient regularization
$$
\int \alpha || \nabla \hat \mu(x) ||_2^2 dx, \qquad \int \beta || \nabla \hat \Lambda(x) ||_2^2 dx.
$$
I feel that there are missing gaps between parametric models vs nonparametric models, though the idea can be understood intuitively.

**Questions:**

See the above section

---

> ### Author Response · Authors · 2023-11-18
>
> Thank you for the comments. We address your concerns below:
>
> > (main) Is there a sharp phase transition? … It would be interesting to calculate, at least under certain simple generative models, the free energy and check if there is a first-order/second-order phase transition.
>
> We agree that the phase transitions are worthy of further investigation. Unfortunately, as seen in the cross sections through $\rho-\gamma$ space, though these phase transitions are continuous, they are not smooth in the order parameter, and we are unable to study this behavior analytically.
>
> > (main) Technically speaking, not sure which part of the paper is innovative---for example, Eqns 7--10, are they new or semi-new (i.e., similar derivations are obtained in a different context)? I would be skeptical that Eqns 7--10 are entirely new. Also, the proposed regularization is well-studied in the literature.
>
> Yes, weight decay is a well-studied method for regularization, but we believe that our parameterization is unique, and splitting the regularization across two separate networks has not yet been studied in depth. [2] also looks at fitting heteroskedastic regression models with different levels of regularization applied to the two networks, and we will add this reference to the camera-ready version.
>
>
> > It is a bit handwaving when transitioning from parameter norm regularization to gradient regularization
>
> We address this concern in the global comment above.
>
>
> [2] Sluijterman, L., Cator, E., & Heskes, T. (2023). Optimal Training of Mean Variance Estimation Neural Networks. arXiv preprint arXiv:2302.08875.

---

### Official Review · Reviewer_yMF5 · 2023-10-31

**Soundness:** 3 good
**Presentation:** 3 good
**Contribution:** 3 good
**Rating:** 8
**Confidence:** 3

**Summary:**

This paper studies heteroskedastic regression problem in the framework of field theory. By modeling parametrized neural network using continuous functions, reparametrizing regularization strength, proposing continuous regularization terms, and approximating the integral over $y$ by a single point, a computationally feasible nonparametric free energy approximating the log likelihood of deep heteroskedastic regression is derived. The reparametrized regularization strength is perceived as order parameters. The field model is solved numerically on a lattice. Abrupt change in the expressiveness of the model and the loss is observed and is interpreted as phase transition. Similar patterns also emerge when using real data and neural networks. The field-theory model implies that one-dimensional hyperparameter searching suffices.

**Strengths:**

A field-theory model is proposed and can explain the pathological behavior of heteroskedastic regression. The model can produce phenomena which appears in regressing tasks with various realistic data sets, indicating the insight obtained from this model is universal to some extent. This makes the results in the paper convincing. The process of deriving the field-theory model is supported by solid reasoning in general, and the experiment is performed using realistic data sets. This paper is well written in general; the figures are informative.

**Weaknesses:**

Although a field-theory model is proposed, little analytical result regarding the phase transition is obtained. There are small issues regarding the writing. I leave the details in Questions.

**Questions:**

* Below equation (7), the authors ‘consider the scenario in which the inner integral is approximated using a single MC sample’. I wonder if there is any justification for this approximation (experiment, argument, reference, etc.).
* I don’t understand the sentence below equation (9): ‘Interestingly, both resulting relationships include a regularization coefficient divided by the density of $x$.’ Does the word ‘regularization coefficient’ refer to the term with Laplace operator, which originates from the regularization term?
* Typo: on top of equation (1), (i.e., $y_i \sim \mathcal{N}(\mu_i, \Lambda_i)$), $\Lambda_i$ or $\Lambda_i^{\frac{1}{2}}$?

---

> ### Author Response · Authors · 2023-11-18
>
> We appreciate the positive feedback and provide replies to your comments below:
>
> > Below equation (7), the authors ‘consider the scenario in which the inner integral is approximated using a single MC sample’. I wonder if there is any justification for this approximation (experiment, argument, reference, etc.).
>
> We address this in the global comment above.
>
> > I don’t understand the sentence below equation (9): ‘Interestingly, both resulting relationships include a regularization coefficient divided by the density of p(x).’ Does the word ‘regularization coefficient’ refer to the term with Laplace operator, which originates from the regularization term?
>
> Yes, that is correct.
>
> > Typo: on top of equation (1)
>
> $\Lambda$ represents the precision (inverse-variance), so $\Lambda^{-1/2}$ is the standard deviation.

---

> > ### Comment · Reviewer_yMF5 · 2023-11-23
> >
> > Thank you for the classification. For the typo, I actually referred to that $\Lambda$ is not used consistently. In the normal distribution three lines above equation (1), $\Lambda$ is used, while in equation (1) and (2), $\Lambda^{-1/2}$ is used.

---

### Official Review · Reviewer_MFfv · 2023-11-01

**Soundness:** 3 good
**Presentation:** 3 good
**Contribution:** 2 fair
**Rating:** 3
**Confidence:** 4

**Summary:**

This paper examines the behaviour of heteroskedastic regression models. By regularizing the model, first with differing levels of weight decay on the mean and covariance functions, and then extending this to the corresponding Dirichlet energies, the authors appeal to tools from statistical mechanics and the calculus of variations in order to derive a system of elliptic partial differential equations that give necessary conditions for energy minimization. This admits a phase diagram in terms of the regularization parameters, describing a two-parameter family of solutions that exhibit phase transitions between different regions of qualitative behaviour. Experimental validation of this behaviour is verified, and the two-dimensional family is reduced to a single dimension for the purposes of hyperparameter optimization.

**Strengths:**

The paper is well written and presented. Drawing insight on tools in machine learning via adjacent fields is always valuable.

**Weaknesses:**

There is a large conceptual leap from the weight decay formulation to using the Dirichlet energy as a regularizer. While the two coincide for linear models, that alone is a tenuous link. Other work has drawn (similarly loose) links to implicit regularization via backwards error analysis of predictive networks trained with SGD, and probably warrants mentioning https://arxiv.org/pdf/2209.13083.pdf.

A single Monte Carlo sample is used in the construction, without further discussion or investigation on the limitations of doing so.

Taking the Dirichlet energy with respect to $p(x)$ may be interesting and warrants discussion (or future work). Assuming $p(x)$ to be uniform for the purposes of numerics is concerning, and doing so may help alleviate this issue.

**Questions:**

Can the authors please address the highlighted weaknesses

**Details Of Ethics Concerns:**

-

---

> ### Author Response · Authors · 2023-11-18
>
> Thank you for the helpful comments and suggestions. We address your concerns above in the global comment (regularization, single MC sample) and below (Dirichlet energy wrt $p(x)$):
>
> > There is a large conceptual leap from the weight decay formulation to using the Dirichlet energy as a regularizer. ... Other work has drawn (similarly loose) links to implicit regularization via backwards error analysis of predictive networks trained with SGD, and probably warrants mentioning https://arxiv.org/pdf/2209.13083.pdf.
>
> Thank you for the useful reference–we will be sure to add it to the camera-ready version. We acknowledge that this was an ad hoc decision, and we will draw in greater ties from the literature. We address this in further detail in the global comment above.
>
> > A single Monte Carlo sample is used in the construction, without further discussion or investigation on the limitations of doing so.
>
> We address this in the global comment above.
>
> > Taking the Dirichlet energy with respect to p(x) may be interesting and warrants discussion (or future work). Assuming p(x) to be uniform for the purposes of numerics is concerning, and doing so may help alleviate this issue.
>
> The analysis we performed is still valid because, pointwise, the limiting cases still hold regardless of the underlying p(x) so long as it is non-zero. For the camera-ready version, we will take the data distribution into account.

---

> > ### Comment · Reviewer_MFfv · 2023-11-22
> >
> > Thanks for your response. While this work has merit, I think that it would benefit from refinement and will be a better contribution at a later conference.

---

### Official Review · Reviewer_fPxx · 2023-11-04

**Soundness:** 1 poor
**Presentation:** 2 fair
**Contribution:** 1 poor
**Rating:** 1
**Confidence:** 4

**Summary:**

This work studies the challenge of conditional variance estimation "from the perspective of statistical physics".  The authors studied the behavior of the regularized learning objective
$$\rho\cdot  E(\log p_N(y\mid \mu(x), \Lambda^{-2}(x))) + (1-\rho)\bigl(\gamma \\|\nabla\mu\\|\_{L_2(P_x)}^2 +  (1-\\gamma) \\|\nabla\Lambda\\|\_{L_2(P_x)}\^2\bigr)$$
where $\mu,sigma$ are the conditional mean and variance functions to be estimated, in the extreme cases of no regularization ($\rho=1$), "no data" ($\rho=0$) and no mean regularization ($\gamma=1$), and presented simulation studies.

**Strengths:**

N/A

**Weaknesses:**

My main concern is that the theoretical results are irrelevant and trivial.
- For example, the "no regularization" regime in Prop. 1 does not describe any reasonable learning algorithms, all of which introduce regularization explicitly or implicitly (e.g. through restrictions to certain function classes, algorithmic regularization through e.g. gradient descent, etc.). If the authors wish to study a nonparametric estimator such as the one defined in their Eq (7), they should impose constraints on the functions (e.g. Sobolev) and carefully choose a rate of vanishing regularization strength *in accordance with the function class*. If the authors wish to study estimators without explicit regularization -- as is common in the analysis of overparameterized models -- they should specify the form of implicit regularization (e.g. gradient descent / gradient flow; model parameterization).
- Furthermore, the challenge of conditional variance estimation arises from overfitting, yet the main result is stated for a *population objective* without any account for sample size.

Additionally, the references to statistical physics appear completely unnecessary.  Calling Eq. (7) a "nonparametric free energy" does not provide any new insight. The proof of the main result also makes no use of techniques or ideas from statistical physics.

**Questions:**

The authors are encouraged to study the proposed learning objective in a relevant and non-trivial regime, and possibly to familiarize themselves with notions in learning theory and nonparametric statistics.

## Post-rebuttal update

I appreciate the authors' response, but it does not address my original concerns, therefore my recommendation remains unchanged.

Regarding the author's question, I have asked for a more precisely formulated asymptotic analysis where the rate of convergence (to 0 or 1) of the regularization hyperparameters are explicitly quantified; this is only possible with at least some concrete conditions imposed to the model family (e.g. DNN models with a certain parameterization).

---

> ### Author Response · Authors · 2023-11-18
>
> Thank you for your comments. We address your concerns below:
>
> > ...the "no regularization" regime in Prop. 1 does not describe any reasonable learning algorithms, all of which introduce regularization explicitly or implicitly
>
> The setting of Proposition 1 is hypothetical in relation to finding a solution to the proposed NFE, which is an abstraction of the typical cross-entropy loss employed in heteroskedastic regression models. The implicit assumptions here are that the model being fit is overparameterized, a universal estimator, twice-differentiable, and is trained until convergence. For these reasons, the only regularization we are considering are the ones we impose through the gradient norm penalties.
>
> > If the authors wish to study a nonparametric estimator such as the one defined in their Eq (7), they should impose constraints on the functions (e.g. Sobolev) and carefully choose a rate of vanishing regularization strength in accordance with the function class.
>
> In summary of our approach, we impose constraints via the gradient norms and take the limits of regularization strengths, $\rho$ and $\gamma$, going to zero and 1 in our analysis. We fear that we may not fully understand your concern. Could you provide some more detail or references to expand on your original comment?
>
> > If the authors wish to study estimators without explicit regularization -- as is common in the analysis of overparameterized models -- they should specify the form of implicit regularization (e.g. gradient descent / gradient flow; model parameterization).
>
> We provide details for training and model parameterization in Appendix B.
>
> > Furthermore, the challenge of conditional variance estimation arises from overfitting, yet the main result is stated for a population objective without any account for sample size.
>
> In our formulation, we can account for sample size through the $\rho$ parameter that scales the importance of the loss between the regularizers and the likelihood.
>
> > Additionally, the references to statistical physics appear completely unnecessary.
>
> Looking at the phase diagrams was an idea inspired by statistical physics. The resulting phase diagram reveals a complex interaction between the regularization strength of the respective functions that cannot be easily summarized by a ratio or linear dependence. This new perspective over combinations of regularization strengths allowed us to notice that an efficient 1-dimensional search along the off-diagonal could be sufficient to find a well-calibrated fit.
>
> > The authors are encouraged to study the proposed learning objective in a relevant and non-trivial regime, and possibly to familiarize themselves with notions in learning theory and nonparametric statistics
>
> We acknowledge that there are other avenues for approaching this problem, but do not consider that a reason to avoid other methods. Learning theory and nonparametric statistics do not relate to phase transitions to our knowledge.

---

### Author Response · Authors · 2023-11-18

We thank the reviewers for their time providing helpful feedback to improve the manuscript.

Among the comments, we noted that there were concerns over the similarity between the gradient norm penalty used in the NFE and the use of an L2 reguarlizer for the neural networks and the choice to use a single MC sample when moving from Equations 7 to 8. We address those concerns here in this global comment and defer responses to the remaining points to the direct replies below.

*Gradient norm vs L2:*
As discussed in [1], there is a correlation between the gradient norm and the L2 penalty. Empirically, we found that an L2 penalty encourages smoothness in a similar way to how the gradient norm does in the NFE.

*Single MC Sample of y|x:*
In practice, with a continuous density in x, we will only have one y value, so taking a single sample of y is a reasonable approximation to make. Furthermore, using a single MC sample is an unbiased estimate of the quantity that we want to maximize.

[1] Dherin, Benoit & Munn, Michael & Rosca, Mihaela & Barrett, David. (2022). Why neural networks find simple solutions: the many regularizers of geometric complexity. 10.48550/arXiv.2209.13083.

---

### Meta-Review · Area_Chair_bVSV · 2023-12-04

**Metareview:**

This paper studies overparameterized heteroskedastic regression models, deriving a energy-based characterization of the learning problem. The authors note a sharp phase transition in modeling behavior based on the choice of regularization parameters, though it is unclear whether this phase transition is significant in practice. Perhaps the major concern of this paper is that the problem setup is not a realistic characterization of overparameterized neural networks, where interpolating solutions are obtained through some implicit regularization or constraints on the function class. There are other questions about the thoretical analysis that could be better addressed in the manuscript, including a sample-size free population objective (unclear how the model is “overparameterized” in this context). The authors should revise the paper to address these questions and submit to a future conference.

**Justification For Why Not Higher Score:**

Many of the reviewers brought up concerns about the model formulation that the authors study in this paper. The authors did not thoroughly assuage these concerns in their rebuttal.

**Justification For Why Not Lower Score:**

N/A

---

### Decision · Program_Chairs · 2024-01-16

Reject